# BETA: Resting-state fMRI Biotypes for tDCS Efficacy in Anxiety Among Older Adults At Risk For Alzheimer's Disease

**Skylar E. Stolte**[1,2,3,*] (ID)          SKYLASTOLTE4444@UFL.EDU

**Junfu Cheng**[4,*] (ID)          JUNFU.CHENG@UFL.EDU

**Chintan Acharya**[1] (ID)          CHINTAN.ACHARYA@UFL.EDU

**Lin Gu**[5] (ID)          RIN.TANI.E8@TOHOKU.AC.JP

**Andrew O'Shea**[3] (ID)          AOSHEA@PHHP.UFL.EDU

**Aprinda Indahlastari**[2,3] (ID)          APRINDA.INDAHLAS@PHHP.UFL.EDU

**Adam J. Woods**[3,6] (ID)          ADAM.WOODS@UTDALLAS.EDU

**Ruogu Fang**[1,3,4,†] (ID)          RUOGU.FANG@BME.UFL.EDU

[1] *J. Crayton Pruitt Family Department of Biomedical Engineering, University of Florida, United States*

[2] *Department of Clinical and Health Psychology, University of Florida, United States*

[3] *Center for Cognitive Aging and Memory, Department of Clinical and Health Psychology, McKnight Brain Institute, University of Florida, USA*

[4] *Department of Electrical and Computer Engineering, University of Florida, United States*

[5] *Research Institute of Electrical Communication, Tohoku University, Japan*

[6] *Behavioral and Brain Sciences, The University of Texas at Dallas, USA*

**Editors:** Accepted for publication at MIDL 2026

## Abstract

Anxiety is usually gauged by self-report, yet a single symptom level can reflect disparate neural circuitry. In Alzheimer's disease and related dementias (ADRD) this heterogeneity becomes a barrier to effective neuromodulation: some patients may benefit from transcranial direct-current stimulation (tDCS), while others may not. To overcome this obstacle, we introduced BETA (Biotypes for tDCS Efficacy in Anxiety), a data-driven pipeline that uses resting-state fMRI functional connectivity to derive anxiety subtypes that are intrinsically linked to tDCS response. A transformer-based variational autoencoder compresses high-dimensional connectivity into a 50-dimensional latent embedding that emphasizes networks implicated in cognitive aging and anxiety. A deep-embedded clustering loss, regularized by a clinically informed term that pulls together individuals who exhibit similar post-tDCS anxiety change, yields four distinct subtypes. Across all subtypes, disrupted coupling between sensory-processing and higher-order cognitive regions emerges as a common hallmark. Crucially, one cluster is resistant to frontal-lobe tDCS, whereas two clusters demonstrate significant anxiety reduction following stimulation. The responsive subtypes are defined by strengthened connectivity between the lateral occipital cortex-superior division (sLOC) and medial frontal cortex (MedFC), and between sLOC and the intracalcarine cortex (ICC). BETA demonstrates that fMRI-based subtyping can directly identify which patients are likely to benefit from tDCS, providing a concrete roadmap for precision psychiatry in ADRD and facilitating tailored therapeutic strategies for anxiety.

**Keywords:** Anxiety, tDCS, ADRD, functional MRI, biotypes

---

[*] Equal Contribution

[†] Corresponding Author

## 1. Introduction

Anxiety is one of the most frequently reported neuropsychiatric symptoms in individuals with Alzheimer's disease (AD) and related dementias (ADRD) (Ferretti et al., 2001; Porter et al., 2003; Patel and Masurkar, 2021). Beyond its symptomatic burden, anxiety has been implicated as both a risk factor for, and a catalyst of, disease progression in AD (Chemerinski et al., 1998; Becker et al., 2018). Consequently, early and precise anxiety interventions may attenuate neurodegeneration and improve quality of life in this vulnerable population.

Current clinical practice relies almost exclusively on self-reported questionnaires, clinical ratings, and historical information to diagnose and monitor anxiety (Penninx et al., 2021). These tools, while essential, are constrained by subjectivity: cultural nuances, individual insight, and context can all skew scores (Evans et al., 2013). Importantly, identical anxiety ratings can mask distinct underlying neurobiological circuits, undermining the fidelity of treatment decision-making (Rao et al., 2023). This phenomenological heterogeneity presents a substantial barrier to the deployment of targeted neuromodulation therapies such as trans-cranial direct-current stimulation (tDCS).

Resting-state functional magnetic resonance imaging (fMRI) offers a means to objectively probe the large-scale connectivity patterns that underpin anxiety. Prior work has identified fMRI-based neuropsychiatric biotypes of depression and anxiety in large adult psychiatric cohorts (Drysdale et al., 2017; Tozzi et al., 2024). However, these studies do not specifically address anxiety phenotypes in older adults or individuals at elevated risk for AD, limiting their applicability to aging and neurodegenerative populations. The emergence of distinct anxiety biotypes could inform which patients are most likely to benefit from neuromodulation, thereby advancing precision psychiatry in AD.

To bridge this gap, we introduce **BETA** (**Biotypes for tDCS Efficacy in Anxiety**), a data-driven framework that integrates resting-state fMRI with a transformer-based variational autoencoder (t-VAE) and a clinical-outcome regularization term. BETA first learns a 50-dimensional latent representation of functional connectivity that captures the canonical networks implicated in cognitive aging and anxiety-namely, the Cingulo-Opercular Network (CON), Frontoparietal Control Network (FPCN), Default Mode Network (DMN), medial frontal cortex, cerebellum, motor cortex, and visual systems-using NeuroSynth-derived masks (Yarkoni et al., 2011). The resulting latent space is then subjected to deep-embedded clustering, iteratively refining cluster assignments while enforcing proximity between participants who exhibit similar anxiety change following frontal-lobe tDCS. This joint optimization yields four distinct anxiety biotypes.

This study is an exploratory, hypothesis-generating analysis of a randomized clinical trial dataset; results are validated through rigorous within-dataset cross-validation and intended to guide future confirmatory trials. Our work is the first to identify biotypes of tDCS efficacy for anxiety, advancing precision neuromodulation in three concise steps:

- **Novel biotype discovery.** Four distinct anxiety biotypes were identified from resting-state fMRI in older adults at risk for Alzheimer's disease—a population that has been substantially underrepresented in prior fMRI-based biotyping and neuromodulation studies—each exhibiting a distinct pattern of response to frontal-lobe tDCS.

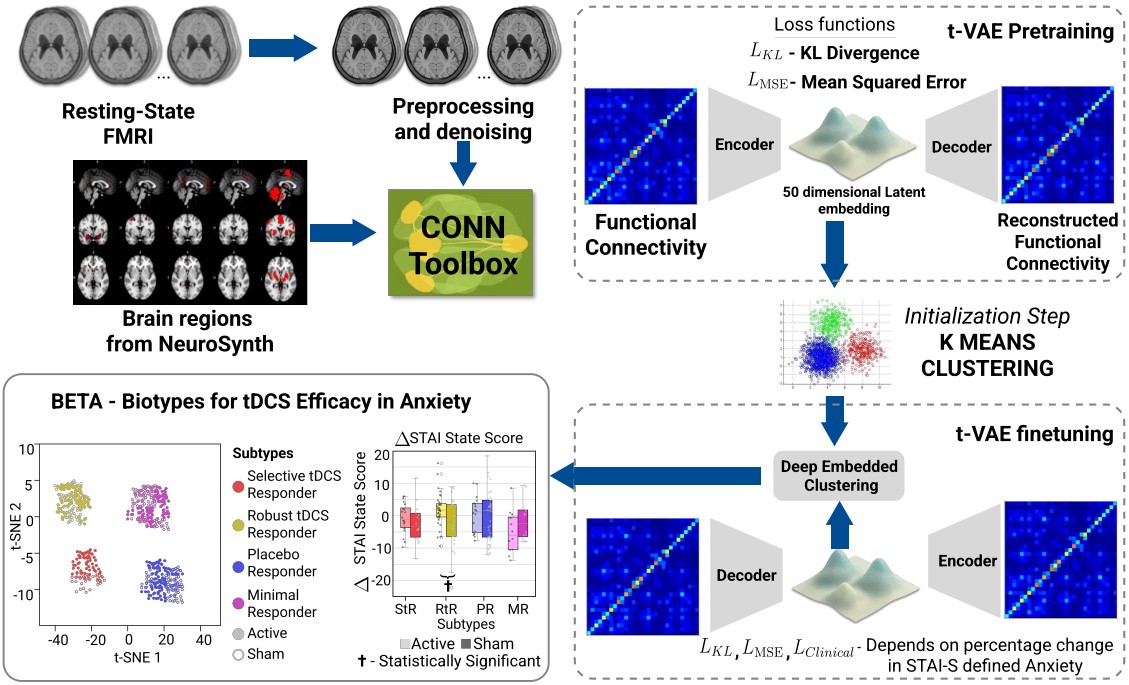

Figure 1: Overview of the BETA pipeline. The BETA pipeline embeds resting-state connectivity into a clinically informed latent space using a transformer-based VAE and deep embedded clustering, yielding distinct anxiety biotypes with differential sham and active tDCS responses.

- **Intervention response prediction.** Two of the biotypes demonstrated marked anxiety relief after tDCS, one showed resistance to tDCS, and the remaining occupied intermediate positions, providing a data-driven way to anticipate treatment outcomes from imaging alone.

- **Clinical decision support.** The biotype assignments were mapped to a simple, actionable tDCS recommendation schema, offering a proof–of–concept framework for individualized neuromodulation in geriatric anxiety.

## 2. Methodology

### 2.1. Dataset

This study used data from a randomized, double-blind trial: The Augmenting Cognitive Training in Older Adults study (ACT; clinicaltrials.gov NCT02851511). ACT examined the impact of transcranial direct-current stimulation (tDCS) combined with cognitive training (CT) on cognitive function in healthy older adults aged 65–89. Exclusion criteria comprised any neurological disorder, cognitive impairment, opportunistic brain infection, major psychiatric illness, unstable or chronic medical condition, MRI contraindication, impaired motor response, GABA-ergic medication use, or left-handedness. The protocol was

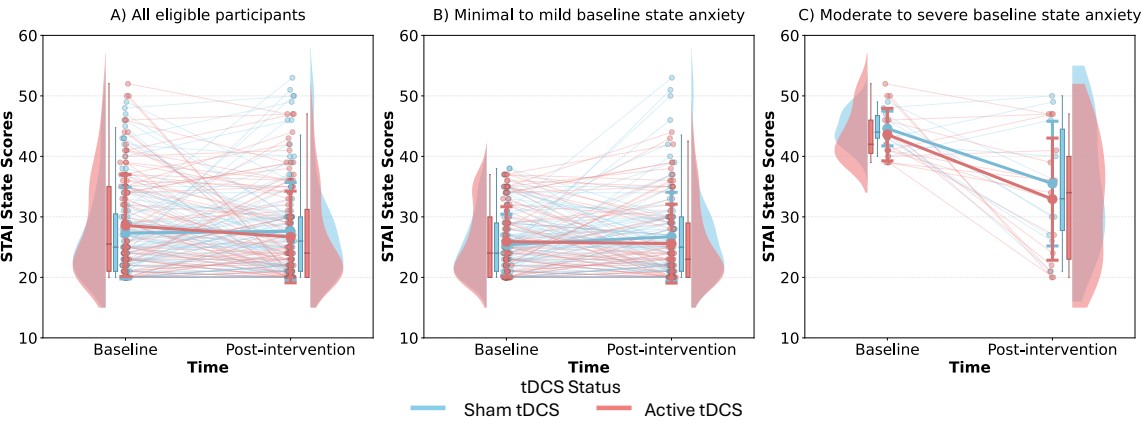

Figure 2: Spaghetti plots of individual State STAI changes from baseline to post-intervention in Sham and Active tDCS groups, stratified by baseline anxiety.

IRB-approved and all participants provided written informed consent. Two research centers performed data acquisition on 3-T Siemens scanners (64-channel head coil at the primary site, 32-channel head coil at the secondary site). Both sites used earplugs to attenuate scanner noise and foam padding to limit head motion. After removing individuals with missing anxiety or fMRI data, the final sample included 199 participants (mean age = $71.27 \pm 4.53$ years; 61 males and 138 females). The sample comprised 99 Sham tDCS and 100 Active tDCS participants.

The State–Trait Anxiety Inventory (STAI) provides a self-report measure of anxiety with 20 items measuring "state" anxiety (STAI–S) and another 20 for "trait" anxiety (STAI–T) (Spielberger et al., 1999), each with a score range of 20 to 80. We collected participants' STAI state score at baseline and post-intervention (12 weeks after screening) time points, and change in State STAI scores for the Sham and Active tDCS groups and the corresponding individual trajectories are visualized in Figure 2.

### 2.1.1. tDCS Intervention

Participants received twelve weeks of tDCS-paired CT. Conventional tDCS was delivered at 2 mA intensity using two $5 \times 7$ cm$^2$ saline-soaked Soterix sponge electrodes over the F3-F4 locations (dorsolateral prefrontal cortex). The tDCS trial was double-blind: active and sham arms received identical setups, and allocation was concealed from both participants and experimenters. Each 40-min CT session began with 20 min of active tDCS. Active stimulation consisted of 2mA for 20 min with 30 s current ramp-up/down; sham stimulation was 2mA for 30 s with identical ramp-up/down, producing the same sensory experience while lacking a biologically meaningful effect.

### 2.1.2. Data Preprocessing and ROI Definition

Resting-state data were preprocessed in CONN (MATLAB 2022b) using standard steps including realignment, slice-timing correction, segmentation, co-registration, normalization

to MNI space, and 8-mm smoothing; one participant was excluded for excessive motion. Regions of interest were defined using NeuroSynth activation maps (Yarkoni et al., 2011) associated with cognitive aging and anxiety (Waner et al., 2023; Takagi et al., 2018), thresholded and binarized to produce data-driven masks spanning Cingulo-Opercular, Frontoparietal Control, Default Mode, medial frontal, cerebellar, motor, and visual networks. These ROIs were used in CONN to extract pairwise functional connectivity (Pearson correlation, Fisher z-transformed).

## 2.2. Deep Embedded Clustering Pipeline

The pipeline compresses each subject's full resting-state functional connectivity (FC) matrix into a latent representation that preserves both anxiety-related network structure and tDCS response, using four tightly integrated stages.

### 2.2.1. Feature extraction with a transformer-based variational auto-encoder (t-VAE)

Variational autoencoders (VAEs) have long been used to summarize high-dimensional fMRI time-series and connectivity maps (Kim et al., 2021; Han et al., 2019; Qiang et al., 2021). Direct clustering on full functional connectivity matrices is challenging due to their high dimensionality, noise, and redundancy. A VAE can provide a way to learn a compact, denoised latent representation that preserves subject-level structure and improves clustering robustness relative to applying standard clustering methods to raw FC features. The VAE can further enable nonlinear representation learning, which is important for capturing distributed and interacting network patterns that are not well described by linear dimensionality reduction.

In this work, we replace the conventional feed-forward encoder with a transformer architecture, allowing the model to learn long-range dependencies that are characteristic of distributed resting-state networks. The encoder comprises four transformer blocks (each with eight self-attention heads and a 64-dimensional hidden state). After the transformer stack a linear projection collapses the concatenated hidden states to a 50-dimensional latent vector $z$. The decoder mirrors this architecture and reconstructs the full functional-connectivity matrix from $z$. We selected a latent dimensionality of 50 based on a nested cross-validation sweep over 20, 50, 80 dimensions; the reconstruction error plateaued at 50, indicating sufficient capacity without over-fitting (Waswani et al., 2017). This latent space serves as the input to the subsequent embedded clustering stage.

### 2.2.2. Initial Clustering

After the transformer-based VAE has produced a 50-dimensional latent representation **z** for each subject, we initialize a set of cluster centroids by running standard $k$-means on the latent vectors. This "warm-start" provides a meaningful segmentation of the data that accelerates convergence and stabilises the optimisation trajectory (Drysdale et al., 2017).

### 2.2.3. Joint Optimization

During training the centroids and the encoder–decoder parameters are updated *jointly* in a single back-propagation loop. The objective function minimises three terms:

i **Reconstruction loss** $\mathcal{L}_{\mathrm{MSE}}$, the mean-squared error between the input FC matrix and the matrix reconstructed from $\mathbf{z}$.

ii **KL divergence** $\mathcal{L}_{\mathrm{KL}}$, which regularises the distribution of $\mathbf{z}$ toward a standard normal prior (Kullback and Leibler, 1951).

iii **Clinical-informed regularisation** $\mathcal{L}_{\mathrm{clinical}}$, a kernel-based penalty that encourages participants with similar changes in the State-Trait Anxiety Inventory (STAI-S) to occupy nearby locations in the latent space (Hofmann et al., 2008). Details of this specially designed loss in this paper is elaborated in Section 2.2.4.

The overall loss is therefore

$$\mathcal{L}_{\mathrm{Total}} = \underbrace{\mathcal{L}_{\mathrm{MSE}}}_{\text{reconstruction error}} + \underbrace{\mathcal{L}_{\mathrm{KL}}}_{\text{KL divergence}} + \beta \underbrace{\mathcal{L}_{\mathrm{Clinical}}}_{\text{clinical-informed loss}} \tag{1}$$

where $\beta$ weights the influence of the clinical term. The weighting coefficient was set to $\beta = 0.1$ based on a brief grid search over $\beta \in \{0.01, 0.05, 0.1, 0.2\}$, balancing preservation of functional connectivity structure with alignment to clinical change. By optimizing the centroids and the VAE parameters simultaneously, the model learns a latent space that is simultaneously faithful to the neuroimaging data and aligned with the observed clinical the response to tDCS.

We regularized the latent space using a KL divergence term to ensure stable joint optimization under multiple objectives, including reconstruction, clustering, and clinical-informed supervision. Without this constraint, the latent representation may fragment or collapse, undermining distance-based clustering. A standard Gaussian prior was adopted as a pragmatic reference rather than a biological assumption, promoting smooth, isotropic latent structure with comparable inter-subject distances, effective Euclidean-based deep embedded clustering, and reduced overfitting to idiosyncratic connectivity patterns.

### 2.2.4. Clinical-Informed Regularization

Given the 12-week intervention, we defined a relative STAI–S change:

$$x_i = \frac{\mathrm{STAI}_{i,\mathrm{post}}}{\mathrm{STAI}_{i,\mathrm{pre}}}, \tag{2}$$

where $\mathrm{STAI}_{i,\mathrm{pre}}$ and $\mathrm{STAI}_{i,\mathrm{post}}$ are the baseline and post-tDCS scores for subject $i$. Participants whose baseline STAI–S exceeds 38 are considered to have moderate/severe state anxiety symptoms and clinically relevant anxiety (CRAL) based on Julian's STAI score interpretation method (Julian, 2011).

For every pair of participants $(i, j)$ in the current mini-batch $B$, we compute a kernel similarity

$$w_{ij} = \exp\left(-\frac{|x_i - x_j|^2}{2\sigma^2}\right), \tag{3}$$

with $\sigma$ fixed to 1.0. We fixed $\sigma = 1$ to match the scale of the normalized relative change in STAI–S, which avoided overly sharp or diffuse clinical weighting and yielded stable optimization. Robustness analyses across brief grid search range of $\beta$ values and modest

variations in $\sigma$ showed stable biotype structure, consistent cluster assignments, and preserved directional patterns of tDCS response, indicating that results were not driven by a single hyperparameter choice. The clinical-informed loss is then the weighted sum of squared latent distances:

$$\mathcal{L}_{\text{clinical}} = \frac{1}{|B|} \sum_{(i,j) \in B} w_{ij} \|\mathbf{z}_i - \mathbf{z}_j\|_2^2, \tag{4}$$

where $\mathbf{z}_i$ denotes the 50-dimensional latent representation of subject $i$. This term penalizes large latent separations between participants who exhibit similar changes in STAI-S, encouraging the embedding to encode treatment efficacy (Hofmann et al., 2008).

The total training objective (Eq. (1)) therefore consists of the reconstruction loss, the KL divergence, and the weighted clinical regulariser, as shown in the previous section.

By optimizing $\mathcal{L}_{\text{clinical}}$ jointly with the VAE and clustering losses, the model learned a latent space that is structured by functional connectivity and selectively relevant to anxiety outcomes following frontal-lobe tDCS, despite the treatment condition (Active vs. Sham) not being used as a supervision signal.

### 2.3. Model Training

Model optimization used Adam (Kingma and Ba, 2014) and followed a two-stage training procedure. In the variational autoencoder (VAE) stage, reconstruction was learned using a loss that combined Mean Squared Error (MSE) with KL divergence to regularize the latent space. The final clustering stage adopted a deep embedded clustering (DEC) objective (Xie et al., 2016), in which KL divergence measures the discrepancy between current cluster assignments and a sharpened target distribution. Throughout training, both active- and sham-tDCS participants were included to ensure that the model distinguished genuine treatment effects from minimal sham-related changes.

In the first stage, the pretraining using only reconstruction and KL divergence losses allowed the model to learn a smooth and well-conditioned latent representation that captures the global structure of connectivity patterns without the instability introduced by clustering constraints. In preliminary experiments, directly optimizing the VAE and clustering objectives from random initialization led to collapsed embeddings and unstable cluster assignments. In the second stage, the pretrained encoder jointly optimized with the deep embedded clustering objective and clinical similarity regularization improved convergence, reproducibility across cross-validation folds, and generalization to held-out participants in testing sets, supporting reliable identification of clinically meaningful anxiety biotypes. Additional methodological detail is provided in the Supplementary Materials.

### 2.4. Model Evaluation

Model evaluation employed a **5-fold cross-validation** framework with no subject overlap between training and testing folds. Within each fold, approximately 80% of subjects were used for training and 20% were held out for testing (roughly 159 training and 40 testing participants per fold). After training, encoder parameters and cluster centroids were frozen, and held-out subjects were projected into the learned latent space. Cross-validation was

used solely to assess the stability of loss convergence and the clinical alignment defined by tDCS efficacy within clusters, not on maximizing separation.

To ensure reproducibility and isolate the effects of the clinical-informed loss, all experiments were conducted with fixed random seeds across NumPy, PyTorch, and scikit-learn.

## 2.5. Inference

During inference, the BETA model operates on a single resting-state fMRI dataset. First, the pre-processed connectivity matrix is fed through the encoder of the pre-trained transformer based VAE, yielding a 50-dimensional latent vector $\mathbf{z}_{\text{new}}$. Next, the Euclidean distance between $\mathbf{z}_{\text{new}}$ and each of the cluster centroids $\{\mathbf{c}_k\}_{k=1}^5$ is computed, and the biotype $k^* = \arg\min_k \|\mathbf{z}_{\text{new}} - \mathbf{c}_k\|_2$ is assigned. Because the STAI-S score is used only during training to weight the clinical-informed loss, it is unnecessary for inference; thus BETA can predict a subject's likelihood of responding to frontal-lobe trans-cranial direct-current stimulation solely from their fMRI-derived latent representation, enabling pre-trial stratification and personalized therapeutic recommendation.

## 2.6. tDCS Efficacy within Clusters

We quantified tDCS efficacy using the pre–post change in STAI-S scores, computed as $\Delta\text{STAI-S} = \text{STAI-S}_{\text{Pre}} - \text{STAI-S}_{\text{Post}}$, where positive values indicate anxiety reduction; this metric is clinically standard, directly interpretable, and preserves meaningful symptom change (Corsaletti et al., 2014). We performed cluster-level pre–post evaluation for each cluster derived from the deep embedded clustering pipeline. We computed $\Delta\text{STAI-S}$ for each participant and acessed normality within each subtype using the Shapiro–Wilk test. Depending on distributional properties, we performed either a one-sample $t$-test or a Wilcoxon signed-rank test. Then, for each of the biotypes we tested whether frontal-lobe tDCS produced a statistically significant change in anxiety relative to sham. Active–Sham differences were tested using parametric and nonparametric two-sample methods (Welch's $t$-test, Mann–Whitney $U$), or permutation testing for small samples with participant number in single digits in three levels of baseline state anxiety separately: (i) the entire cluster samples $(n)$, (ii) participants with moderate or severe state anxiety (STAI-S > 38), and (iii) participants with minimal or mild state anxiety (STAI-S ≤ 38).

## 3. Experiments and Results

### 3.1. Cluster Number and Excluded Cluster

The number of clusters was set to $k = 5$ for interpretability and statistical power, consistent with prior neuropsychiatric biotyping work showing that modest $k$ values yield stable, clinically meaningful subtypes.

In preliminary analyses, smaller values ($k = 3$–4) produced overly broad groupings that merged divergent connectivity patterns and treatment responses, whereas larger values ($k \geq 6$) yielded multiple very small, poorly interpretable clusters. Across cross-validation folds, $k = 5$ initialization produced consistent centroids and stable convergence during joint VAE–clustering optimization, yielding four clusters with adequate sample sizes and one small cluster ($n = 3$). The small cluster exhibited highly idiosyncratic connectivity patterns

and heterogeneous clinical trajectories, precluding reliable estimation of pre–post anxiety changes or Active–Sham differences. Because statistical testing would be underpowered and potentially misleading, this cluster was excluded from all inferential analyses. Its exclusion did not affect the composition, connectivity profiles, or treatment-response patterns of the remaining four clusters.

### 3.2. tDCS Efficacy within Clusters

We identified four clusters using our BETA pipeline after removing one small cluster with $n = 3$ due to very limited size, which will lead to lack of sufficient statistical power for meaningful inference. The remaining four clusters: Selective tDCS Responder, Robust tDCS Responder, Placebo Responder, and Minimal Responder subtypes—were retained for all subsequent analyses. The demographic and psychological profiles of these four clusters are presented in the supplementary material. Cluster level pre–post evaluation and Active–Sham difference evaluation results was summarized in as illustrated in Table S1.

Pre–post analyses revealed differential STAI-S changes across clusters. Participants with moderate to severe baseline state anxiety in the Selective tDCS Responder and Placebo Responder clusters showed statistically significant STAI-S reductions, whereas the Minimal Responder and Robust tDCS Responder clusters showed no significant change. The Selective tDCS Responder cluster exhibited the largest effect, with a mean STAI-S reduction of $26.0 \pm 6.08$ under active tDCS ($n = 3$). This reduction exceeds the minimal important difference of 10 points proposed by Corsaletti et al. (Corsaletti et al., 2014), indicating a clinically meaningful improvement.

In Active–Sham difference evaluation across the entire samples in four clusters, the Robust tDCS Responder subtype cluster showed statistically significant ($p = 0.02$) superior state anxiety symptoms reduction under Active stimulation with tDCS. In addition, the minimal/mild baseline anxiety subgroup of the Robust tDCS Responder subtype also shows a statistically significant ($p = 0.03$) Active–Sham advantage among participants confirming superior improvements under Active stimulation. In contrast, the Selective tDCS Responder and Minimal Responder subtypes showed no statistically significant Active–Sham differences. The Placebo Responder subtype exhibited larger mean STAI-S reductions under sham stimulation, though this effect was not statistically significant. Full Active–Sham comparisons are reported in the Supplementary Material.

### 3.3. Cluster-level differences in Functional Connectivity

Figure 3 presents the four functional connections that differ most significantly from the population mean for each biotype. Across the four connectivity-based subtypes, the most notable finding is that active tDCS produced a meaningful reduction in anxiety only in groups showing stronger networks linking the lateral occipital cortex, angular gyrus and frontal control regions—including the superior frontal gyrus, inferior frontal gyrus, middle frontal gyrus and precentral gyrus. These enhanced circuits characterized the Selective tDCS Responder and Robust tDCS Responder subtypes. Notably, the Robust tDCS Responder subtype was the only group to show a significant overall improvement with active tDCS compared to sham tDCS. The Selective tDCS Responder group showed some common features in the functional connectivity trends. While it did not reach statistical significance,

Table 1: Pre–post change in State–Trait Anxiety Inventory–State (STAI-S) scores ($\Delta$STAI-S), stratified by baseline STAI-S severity, tDCS condition, and data-driven subtype. Values are reported as mean (SD), with sample sizes shown as $n$. Asterisks (*) denote statistically significant within-cluster pre–post changes ($p < .05$). Braces labeled with † indicate subtypes exhibiting a significant Active vs. Sham difference in STAI-S improvement ($p < .05$). Bolded values reflect reductions exceeding the minimal important difference of 10 points, indicating clinically meaningful improvement.

| Subtype | Selective tDCS Responder (n=39) | Robust tDCS Responder (n=68) | Placebo Responder (n=60) | Minimal Responder (n=29) |
|---|---|---|---|---|
| Minimal/Mild Sham tDCS | -0.5 (6.50) $n = 14$ | -2.7 (7.43) $n = 29$ | 0.9 (7.99) $n = 32$ | -4.0 (8.47) $n = 12$ |
| Minimal/Mild Active tDCS | 0.1 (5.56) $n = 20$ | 1.8 (5.82) $n = 34$ | -0.9 (5.73) $n = 18$ | -1.2 (6.58) $n = 12$ |
| Moderate/Severe Sham tDCS | **15.0** (2.83) $n = 2$ | N/A $n = 0$ | **12.8** (13.00) $n = 4$ | 2.5 (9.57) $n = 4$ |
| Moderate/Severe Active tDCS | **26.0** (6.08)* $n = 3$ | 7.0 (11.47) $n = 5$ | 7.3 (6.95)* $n = 6$ | 3.0 (N/A) $n = 1$ |
| All Sham tDCS | 1.4 (8.07) $n = 16$ | -2.7 (7.43) $n = 29$ | 2.2 (9.24) $n = 36$ | -2.4 (8.91) $n = 16$ |
| All Active tDCS | 3.5 (10.47) $n = 23$ | 2.5 (6.81) $n = 39$ | 1.2 (6.93) $n = 24$ | -0.8 (6.40) $n = 13$ |

there was a clinically meaningful (greater than 10-point) difference in active vs. sham tDCS in people with moderate/severe anxiety. Hence, the similarities in functional trends between these two groups may be indicative of brain functional connections that are more susceptible to improvement from tDCS intervention. Generally, hyperactive engagement within these networks was associated with tDCS responsiveness. These findings underscore that targeted tDCS interventions may be particularly effective for specific connectivity-based subtypes of anxiety.

## 3.4. Ablation Analysis

We conducted an ablation analysis to assess the value of the clinical-informed loss term ($L_{\text{Clinical}}$). Models trained with and without this term were compared based on cluster-wise differences in anxiety change. The full model (with $L_{\text{Clinical}}$) yielded significant cluster differences (Welch's ANOVA: $p = 0.02$), whereas the ablation model showed no significant separation ($p = 0.10$). The full model demonstrated significant tDCS responses in the Robust tDCS Responder subtype and clinically meaningful anxiety reductions in the Selective tDCS Responder subtype, whereas the ablation model showed no significant group differences in tDCS outcomes. These results show that the clinical-informed loss is crucial for shaping latent representations that capture differential tDCS responses. Without it, the t-VAE encodes general anxiety features, but adding $L_{\text{Clinical}}$ guides the model toward representations better suited for analyzing targeted clinical interventions.

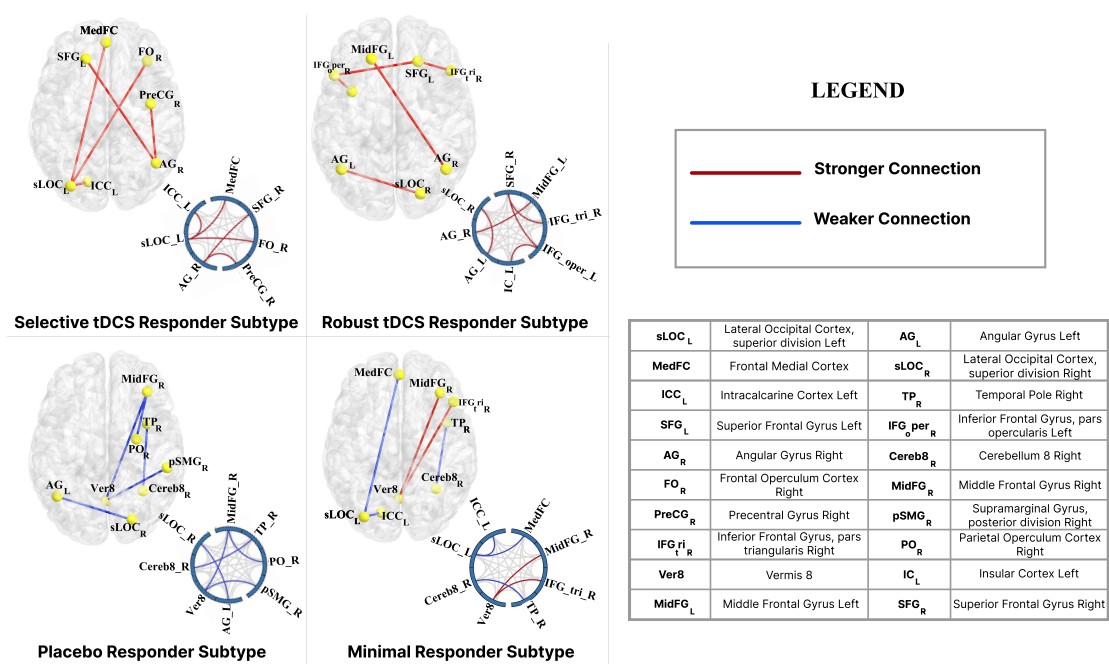

Figure 3: Distinct hyper- and hypo-connectivity patterns across the four biotypes. Blue lines denote weaker-than-average and red lines stronger-than-average connections, with effect size defined as deviation from the cohort mean functional connectivity. Quantitative measures are reported in the Supplementary Material.

## 4. Discussions

This work identified fMRI-derived biomarkers of anxiety in older adults. Our findings implicate regions such as the lateral occipital cortex (LOC) and the angular gyrus (AG) in anxiety. Previous research has shown that anxiety is often characterized by impaired functioning between regions involved in sensory processing (e.g., LOC,AG) and higher cognitive control (e.g. gyri in prefrontal regions) (Langhammer et al., 2024). The LOC is associated with visual processing (Li et al., 2020). Most anxiety research has associated hyperactivity in the LOC with anxiety (Langhammer et al., 2024). The most frequently observed connection across clusters was between the LOC and the medial frontal cortex. The mechanism may involve heightened fear processing and response (Li et al., 2020). However, our results suggest differential anxiety biotypes associated with reduced activity in the lateral occipital cortex. This reduced activity may contribute to anxiety through impaired processing of environmental stimuli (van Dam and Chrysikou, 2021). The current study indicates that increased LOC activity is associated with greater anxiety improvement following tDCS. These findings suggest that tDCS may enhance inhibitory control in this region. Further investigation is needed to understand how reduced activity in this region contributes to anxiety so that interventions can be improved for these individuals. Another main area of interest is the AG, particularly its connection to the medial frontal gyrus. This connection has potential impacts on attention, memory, and language skills (De Boer et al., 2020).

tDCS appears more effective in individuals with hyperactivity in this region than in those with hypoactivity. Previous research has focused on the AG as a target for tDCS to improve cognitive abilities related to dementia and memory (Hu et al., 2022). Our observations in the current study indicate that tDCS may improve cognitive performance related to the angular gyrus by helping regulate attention.

This study was limited by the dataset size. The original data source was based on relatively healthy older adults at risk of AD; it was not an anxiety dataset. Some clustering groups did not include enough data to compare individuals who had clinically significant anxiety across active and sham tDCS. For example, the Robust tDCS Responder subtype cluster did not include individuals with clinically relevant anxiety in the sham tDCS condition. On average, 10–20% of participants in each group exhibited clinically relevant anxiety.

## 5. Conclusions

This study demonstrates that resting-state fMRI connectivity can stratify individuals into biotypes with distinct responses to tDCS for anxiety. Our BETA pipeline processes new fMRI data to generate personalized intervention recommendations, allowing clinicians to target tDCS more effectively according to neuroimaging-derived subtypes and initial anxiety level. Participants in the Robust tDCS Responder subtype may benefit from active tDCS against anxiety, while the Selective tDCS Responder shows the strongest clinical response. In contrast, the Minimal and Placebo Responder subtypes show little or no improvement, highlighting the value of biotype assignment for targeting and prioritizing interventions.

For future work, we aim to broaden the framework to include diverse participant profiles, additional clinical settings, and alternative neuromodulation modalities. The present work offers a foundational step toward precision psychiatry, underscoring the value of individualized, connectivity-based recommendations in optimizing anxiety interventions.

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

## Supplementary Materials

### S1. fMRI Data Acquisition

Neuroimaging data were acquired on harmonized 3 T Siemens scanners across two sites. Data were collected on a Siemens Magnetom Prisma scanner with a 64-channel head coil and a Siemens Magnetom Skyra scanner with a 32-channel head coil. Earplugs and foam padding were used to reduce noise and head motion.

Functional images were acquired using a gradient-echo EPI BOLD sequence (TR = 1800 ms, TE = 2.26 ms, voxel size = 1.0 mm isotropic, FOV = 256 × 256 × 176 mm). Inter-site consistency was monitored using regular FBIRN, GABA, and human phantom scans. The mean signal-to-noise ratio across sites was 12.1 ± 1.54.

### S2. Additional Dataset Details

The State-Trait Anxiety Inventory (STAI) has two subscales, each with a score range of 20 to 80. Change in STAI-S for the Sham and Active tDCS groups are summarized in Supplementary Table S1.

Table S1: Sham and Active tDCS State STAI score change from pre-intervention to post-intervention. $n$: number of participants, $STAI$-$S$: STAI state score, $SD$: Standard deviation, $IQR$: Interquartile range

| Measure | Group | n | Mean | Median | SD | IQR |
|---------|-------|---|------|--------|-----|-----|
| Baseline STAI-S | Sham tDCS | 99 | 27.33 | 25 | 7.58 | 9.50 |
| Baseline STAI-S | Active tDCS | 100 | 28.57 | 25 | 8.43 | 14.00 |
| Baseline STAI-S | All | 199 | 27.95 | 25 | 8.02 | 12.50 |
| Post STAI-S | Sham tDCS | 99 | 27.62 | 26 | 8.07 | 9.00 |
| Post STAI-S | Active tDCS | 100 | 26.68 | 24 | 7.58 | 11.25 |
| Post STAI-S | All | 199 | 27.15 | 25 | 7.82 | 9.00 |
| STAI-S (Pre-Post) | Sham tDCS | 99 | −0.28 | 0 | 8.64 | 5.50 |
| STAI-S (Pre-Post) | Active tDCS | 100 | 1.89 | 0 | 7.77 | 7.00 |
| STAI-S (Pre-Post) | All | 199 | 0.81 | 0 | 8.27 | 6.00 |

### S3. NeuroSynth-Based ROI Construction

Anxiety-related regions of interest (ROIs) were defined using meta-analytic data from NeuroSynth. ROIs were derived from *association maps* rather than uniformity maps, as association maps identify brain regions selectively associated with a given term while controlling for base-rate activation across the literature, providing greater specificity for disorder-relevant network definition. The NeuroSynth query term *"anxiety"* was submitted via the NeuroSynth web interface, producing an association map based on an automated meta-analysis of 449 anxiety-related studies.

The association map was thresholded using the default NeuroSynth web interface options, with the threshold interactively adjusted to retain suprathreshold voxels showing positive association with anxiety. The thresholded map was binarized, and spatially contiguous suprathreshold clusters were extracted, with each cluster defined as a distinct ROI. Because ROI generation was performed via the graphical interface rather than a fully scripted pipeline, the exact numeric threshold value was not explicitly recorded; however, all ROIs were fully specified by their voxel extents and spatial coordinates. This procedure yielded 182 ROIs, which were assigned to canonical functional networks (limbic, frontal control, default mode, sensory, and subcortical) based on anatomical location and established network definitions. A complete list of NeuroSynth-derived ROIs, including ROI name, network assignment, voxel count, and MNI centroid coordinates, is provided in Supplementary Table S2.

Table S2: NeuroSynth-derived regions of interest (ROIs) and network assignments.

| ROI Name | Network |
| --- | --- |
| atlas.Cuneal Cortex (Right) | Atlas ROI |
| atlas.Cuneal Cortex (Left) | Atlas ROI |
| atlas.Supracalcarine Cortex (Left) | Atlas ROI |
| atlas.Supracalcarine Cortex (Right) | Atlas ROI |
| atlas.Intracalcarine Cortex (Left) | Atlas ROI |
| atlas.Intracalcarine Cortex (Right) | Atlas ROI |
| atlas.Lingual Gyrus (Right) | Atlas ROI |
| atlas.Lingual Gyrus (Left) | Atlas ROI |
| networks.Visual.Medial (2, -79, 12) | Network |
| Yeo.Visual | Network |
| atlas.Occipital Fusiform Gyrus (Left) | Atlas ROI |
| atlas.Occipital Fusiform Gyrus (Right) | Atlas ROI |
| atlas.Occipital Pole (Right) | Atlas ROI |
| atlas.Occipital Pole (Left) | Atlas ROI |
| networks.Visual.Occipital (0, -93, -4) | Network |
| atlas.Lateral Occipital Cortex, inferior division (Right) | Atlas ROI |
| networks.Visual.Lateral (Right) (38, -72, 13) | Network |
| networks.Visual.Lateral (Left) (-37, -79, 10) | Network |
| atlas.Lateral Occipital Cortex, inferior division (Left) | Atlas ROI |
| atlas.Temporal Occipital Fusiform Cortex (Left) | Atlas ROI |
| atlas.Temporal Occipital Fusiform Cortex (Right) | Atlas ROI |
| atlas.Inferior Temporal Gyrus, temporo-occipital part (Right) | Atlas ROI |
| atlas.Inferior Temporal Gyrus, temporo-occipital part (Left) | Atlas ROI |
| RestingState | Concept / Task |

*Continued on next page*

| ROI Name | Network |
| --- | --- |
| Frontoparietal | Concept / Task |
| Attention | Concept / Task |
| Yeo.Dorsal Attention | Network |
| atlas.Superior Parietal Lobule (Right) | Atlas ROI |
| networks.DorsalAttention.IPS (Right) (39, -42, 54) | Network |
| atlas.Superior Parietal Lobule (Left) | Atlas ROI |
| networks.DorsalAttention.IPS (Left) (-39, -43, 52) | Network |
| atlas.Supramarginal Gyrus, anterior division (Left) | Atlas ROI |
| networks.Salience.SMG (Left) (-60, -39, 31) | Network |
| atlas.Supramarginal Gyrus, anterior division (Right) | Atlas ROI |
| networks.Salience.SMG (Right) (62, -35, 32) | Network |
| atlas.Supramarginal Gyrus, posterior division (Right) | Atlas ROI |
| atlas.Inferior Frontal Gyrus, pars opercularis (Right) | Atlas ROI |
| atlas.Frontal Operculum Cortex (Right) | Atlas ROI |
| networks.Salience.AInsula (Right) (47, 14, 0) | Network |
| networks.Salience.AInsula (Left) (-44, 13, 1) | Network |
| Yeo.Salience | Network |
| atlas.Parietal Operculum Cortex (Left) | Atlas ROI |
| atlas.Central Opercular Cortex (Left) | Atlas ROI |
| atlas.Insular Cortex (Left) | Atlas ROI |
| atlas.Insular Cortex (Right) | Atlas ROI |
| atlas.Planum Temporale (Right) | Atlas ROI |
| atlas.Parietal Operculum Cortex (Right) | Atlas ROI |
| atlas.Central Opercular Cortex (Right) | Atlas ROI |
| networks.Sensorimotor.Lateral (Right) (56, -10, 29) | Network |
| Yeo.Somatosensory | Network |
| networks.Sensorimotor.Lateral (Left) (-55, -12, 29) | Network |
| atlas.Precentral Gyrus (Left) | Atlas ROI |
| atlas.Postcentral Gyrus (Left) | Atlas ROI |
| networks.Sensorimotor.Superior (0, -31, 67) | Network |
| atlas.Postcentral Gyrus (Right) | Atlas ROI |
| atlas.Precentral Gyrus (Right) | Atlas ROI |
| atlas.Supplementary Motor Area (Right) | Atlas ROI |
| atlas.Supplementary Motor Area (Left) | Atlas ROI |
| networks.DorsalAttention.FEF (Left) (-27, -9, 64) | Network |
| networks.DorsalAttention.FEF (Right) (30, -6, 64) | Network |

| ROI Name | Network |
|---|---|
| networks.Salience.RPFC (Right) (32, 46, 27) | Network |
| networks.Salience.RPFC (Left) (-32, 45, 27) | Network |
| networks.Salience.ACC (0, 22, 35) | Network |
| atlas.Anterior Cingulate Cortex | Atlas ROI |
| atlas.Putamen (Right) | Atlas ROI |
| atlas.Putamen (Left) | Atlas ROI |
| atlas.Thalamus (Left) | Atlas ROI |
| atlas.Thalamus (Right) | Atlas ROI |
| atlas.Nucleus Accumbens (Left) | Atlas ROI |
| atlas.Nucleus Accumbens (Right) | Atlas ROI |
| salience | Concept / Task |
| atlas.Pallidum (Right) | Atlas ROI |
| atlas.Pallidum (Left) | Atlas ROI |
| Sad | Concept / Task |
| threatening | Concept / Task |
| Reward | Concept / Task |
| MonetaryReward | Concept / Task |
| Default | Concept / Task |
| Yeo.Default | Network |
| Cognitive Control | Concept / Task |
| Yeo.Frontoparietal | Network |
| atlas.Precuneus | Atlas ROI |

## S4. Model Architecture

We employed a Transformer-based variational autoencoder (VAE) to learn a compact and structured latent representation of whole-brain functional connectivity patterns. The model followed the standard encoder–decoder VAE framework, with self-attention mechanisms replacing conventional convolutional or fully connected hidden layers in order to capture global dependencies among connectivity features.

### S4.1. Input Representation

For each participant, the functional connectivity matrix was vectorized into a one-dimensional feature vector of length $D$, corresponding to the number of unique connectivity edges. This vector served as the direct input to the VAE. Let $\mathbf{x} \in \mathbb{R}^D$ denote the input connectivity feature vector.

### S4.2. Encoder

The encoder began with a linear embedding layer that projects the input vector $\mathbf{x}$ into a 512-dimensional feature space. The embedded representation was then processed by a Transformer encoder composed of two stacked Transformer encoder layers, each using four self-attention heads and a feedforward hidden dimension of 512. To accommodate the

Transformer architecture, the embedded vector was treated as a single-token sequence by introducing a singleton sequence dimension.

The output of the Transformer encoder was subsequently passed through two parallel fully connected layers to estimate the parameters of the approximate posterior distribution:

$$q_\phi(\mathbf{z} \mid \mathbf{x}) = \mathcal{N}(\boldsymbol{\mu}, \mathrm{diag}(\boldsymbol{\sigma}^2)),$$

where $\boldsymbol{\mu} \in \mathbb{R}^{50}$ and $\log \boldsymbol{\sigma}^2 \in \mathbb{R}^{50}$ denote the latent mean and log-variance vectors, respectively. The latent dimensionality was fixed at 50 to balance representational capacity and regularization.

### S4.3. Latent Sampling

Latent variables were sampled using the reparameterization trick to enable gradient-based optimization:

$$\mathbf{z} = \boldsymbol{\mu} + \boldsymbol{\epsilon} \odot \exp\left(\tfrac{1}{2} \log \boldsymbol{\sigma}^2\right), \quad \boldsymbol{\epsilon} \sim \mathcal{N}(\mathbf{0}, \mathbf{I}).$$

### S4.4. Decoder

The decoder mirrors the encoder architecture. The latent vector $\mathbf{z}$ was first linearly projected into the same 512-dimensional embedding space and then processed by a two-layer Transformer encoder block (used here as a decoder with self-attention). A final linear output layer mapped the Transformer output back to the original input dimension $D$, producing a reconstruction $\hat{\mathbf{x}} \in \mathbb{R}^D$ of the input connectivity features.

## S5. Model Training Details

During optimization, we used the Adam optimizer with a learning rate of $1 \times 10^{-5}$ for a maximum of 200 epochs. Mini-batch training was performed with a batch size of 64 subjects. An initial warmup learning scheduler was used for 5 epochs to ensure a stable gradient during the beginning of the learning process. The model training loss balanced terms for KL Divergence and Mean Squared Error (MSE). The initial variational autoencoder stage learned how to reconstruct the functional correlation data. The final clustering model learned the optimal clustering using a clustering objective that was based off of deep embedded clustering approaches (Xie et al., 2016).

During both training schemes, the MSE loss refers to the ability to reconstruct the input data. The Variational Autoencoder applies KL divergence as the difference between the latent space and the normal distribution. The Deep Embedded Clustering using KL divergence as the difference between the current clustering assignments and a sharpened version of the clusters, as in (Xie et al., 2016). In addition to the standard reconstruction and KL divergence losses, the VAE objective incorporated a clinical similarity regularization term that penalized squared Euclidean distances between latent representations weighted by pairwise similarity in clinical anxiety measures, encouraging clinically meaningful organization of the latent space.

Importantly, both active-tDCS and sham-tDCS participants were retained throughout training so that the model learned to distinguish genuine treatment effects from the negligible changes seen in the sham group. Consistent with best practices in data-driven clustering,

the excluded cluster was interpreted as likely reflecting outlier subjects or rare connectivity configurations rather than a stable or generalizable anxiety biotype. All primary conclusions were therefore based on the four retained clusters, which together account for the vast majority of participants and demonstrate robust, interpretable differences in functional connectivity and tDCS response.

## S6. Group-level Difference in Functional Connectivity

This experiment extracted the functional connectivity networks that were statistically different for each cluster compared to the population averages. The key outcome of this analysis was to learn how anxiety presents differently in each cluster. This information could provide potential insight into why the clusters may respond differently to tDCS interventions. We used one-sample t-testing to extract the functional connectivity differences that were significant for each cluster compared to the entire population. The pipeline used many one-sample t-tests for each cluster and functional connectivity network, so the false discovery rate (FDR) method (Benjamini and Hochberg, 1995) was applied to correct for multiple comparisons.

Table S3 provides the corresponding quantitative details, including the mean connectivity value for each connection within every biotype and the 95% confidence interval for the population average. Together, these results demonstrate that each biotype exhibits a unique functional connectivity signature, highlighting the neurobiological heterogeneity across groups.

Table S3: Average functional connectivity strengths across four subtypes corresponding four clusters. ↑ and ↓ mean significantly stronger and significantly weaker connections for the given cluster, respectively. ROI Abbreviations and Full Region Names were listed in Table S4. The impact of tDCS across all participant in cluster: * = clinically significant anxiety improved with tDCS, ** = all anxiety improved with tDCS.

| Subtype | Selective tDCS Responder$^*$ | Robust tDCS Responder$^{**}$ | Placebo Responder | Minimal Responder | All Participants |
|---|---|---|---|---|---|
| $sLOC_L - MedFC$ | $0.399 \uparrow$ | $0.218$ | $0.161 \downarrow$ | $0.0930 \downarrow$ | $0.219 \pm 0.0380$ |
| $Ver8 - MidFG_R$ | $-0.0136 \downarrow$ | $0.0220$ | $-0.0381 \downarrow$ | $0.197 \uparrow$ | $0.0216 \pm 0.0290$ |
| $IFG_{tri_R} - Ver8$ | $0.0332$ | $0.0111$ | $-0.0153 \downarrow$ | $0.197 \uparrow$ | $0.0301 \pm 0.0310$ |
| $TP_R - Cereb8_R$ | $0.171 \uparrow$ | $0.0885$ | $0.0679$ | $-0.0672 \downarrow$ | $0.0783 \pm 0.0310$ |
| $sLOC_L - ICC_L$ | $0.371 \uparrow$ | $0.206$ | $0.181 \downarrow$ | $0.118 \downarrow$ | $0.217 \pm 0.0350$ |
| $SFG_L - AG_R$ | $0.461 \uparrow$ | $0.450 \uparrow$ | $0.317 \downarrow$ | $0.404$ | $0.398 \pm 0.0340$ |
| $sLOC_L - FO_R$ | $0.264 \uparrow$ | $0.155$ | $0.127$ | $0.0533 \downarrow$ | $0.149 \pm 0.0320$ |
| $AG_R - PreCG_R$ | $0.407 \uparrow$ | $0.361 \uparrow$ | $0.246 \downarrow$ | $0.323$ | $0.325 \pm 0.0320$ |
| $MidFG_L - AG_R$ | $0.364 \uparrow$ | $0.384 \uparrow$ | $0.230 \downarrow$ | $0.254 \downarrow$ | $0.308 \pm 0.0360$ |
| $IFG_{oper_L} - Cereb8_R$ | $0.214 \uparrow$ | $0.0975$ | $0.0628$ | $-0.0225 \downarrow$ | $0.0949 \pm 0.0330$ |
| $AG_L - sLOC_R$ | $0.635 \uparrow$ | $0.661 \uparrow$ | $0.515 \downarrow$ | $0.477 \downarrow$ | $0.580 \pm 0.0370$ |
| $IC_L - IFG_{oper_L}$ | $0.557 \uparrow$ | $0.574 \uparrow$ | $0.445 \downarrow$ | $0.499$ | $0.513 \pm 0.0350$ |
| $SFG_R - IFG_{oper_L}$ | $0.655 \uparrow$ | $0.668 \uparrow$ | $0.517 \downarrow$ | $0.530 \downarrow$ | $0.592 \pm 0.0400$ |
| $SFG_R - IFG_{tri_R}$ | $0.714$ | $0.729 \uparrow$ | $0.645$ | $0.640 \downarrow$ | $0.678 \pm 0.0380$ |
| $Ver8 - pSMG_R$ | $0.0243$ | $0.0319$ | $0.00341 \downarrow$ | $0.191 \uparrow$ | $0.0453 \pm 0.0280$ |
| $MidFG_R - PO_R$ | $-0.0702$ | $-0.0340 \uparrow$ | $-0.160 \downarrow$ | $-0.103$ | $-0.0842 \pm 0.0320$ |

Table S4: ROI Abbreviations and Full Region Names

| Abbrev. | Region Name |
|---------|-------------|
| $\text{sLOC}_{\text{L}}$ | Lateral Occipital Cortex, superior division Left |
| MedFC | Frontal Medial Cortex |
| $\text{ICC}_{\text{L}}$ | Intracalcarine Cortex Left |
| $\text{SFG}_{\text{L}}$ | Superior Frontal Gyrus Left |
| $\text{AG}_{\text{R}}$ | Angular Gyrus Right |
| $\text{FO}_{\text{R}}$ | Frontal Operculum Cortex Right |
| $\text{PreCG}_{\text{R}}$ | Precentral Gyrus Right |
| $\text{IFG}_{\text{tri R}}$ | Inferior Frontal Gyrus, pars triangularis Right |
| Ver8 | Vermis 8 |
| $\text{MidFG}_{\text{L}}$ | Middle Frontal Gyrus Left |
| $\text{AG}_{\text{L}}$ | Angular Gyrus Left |
| $\text{sLOC}_{\text{R}}$ | Lateral Occipital Cortex, superior division Right |
| $\text{TP}_{\text{R}}$ | Temporal Pole Right |
| $\text{IFG}_{\text{oper R}}$ | Inferior Frontal Gyrus, pars opercularis Right |
| $\text{Cereb8}_{\text{R}}$ | Cerebellum 8 Right |
| $\text{MidFG}_{\text{R}}$ | Middle Frontal Gyrus Right |
| $\text{pSMG}_{\text{R}}$ | Supramarginal Gyrus, posterior division Right |
| $\text{PO}_{\text{R}}$ | Parietal Operculum Cortex Right |
| $\text{IC}_{\text{L}}$ | Insular Cortex Left |
| $\text{SFG}_{\text{R}}$ | Superior Frontal Gyrus Right |

## S7. Cluster-by-Cluster Results Across Baseline Severity Levels

In cluster level pre–post evaluation, for each cluster, change scores were tested against zero using one-sample $t$-tests when Shapiro–Wilk normality was not violated ($p < 0.05$) and Wilcoxon signed-rank tests otherwise. Subgroups in each cluster defined by baseline STAI S in clusters with $n = 1$ or $n = 0$ were not tested. For the Minimal/Mild baseline anxiety group, Active–Sham differences were evaluated using Welch's two-sample $t$-test (unequal variances) and the nonparametric Mann–Whitney $U$ test, accompanied by assumption checks via Shapiro–Wilk normality tests and Levene's median-based variance test. Effect sizes were computed using Cohen's $d$, Hedges' $g$, and Glass's $\Delta$. Because sample sizes were limited in the Moderate/Severe baseline group, condition differences were assessed using a two-sample permutation test on mean differences (10,000 permutations, two-sided), supplemented by the same normality and variance diagnostics and effect size estimates. The permutation test evaluates whether the observed Active–Sham difference could arise by chance by repeatedly shuffling group labels and recomputing the mean difference across thousands of random reallocations. Together, this approach ensured that each comparison was evaluated with statistical methods appropriate for the underlying sample size, distribution, and variance structure. The change of STAI-S across each cluster was illustrated in Figure S1.

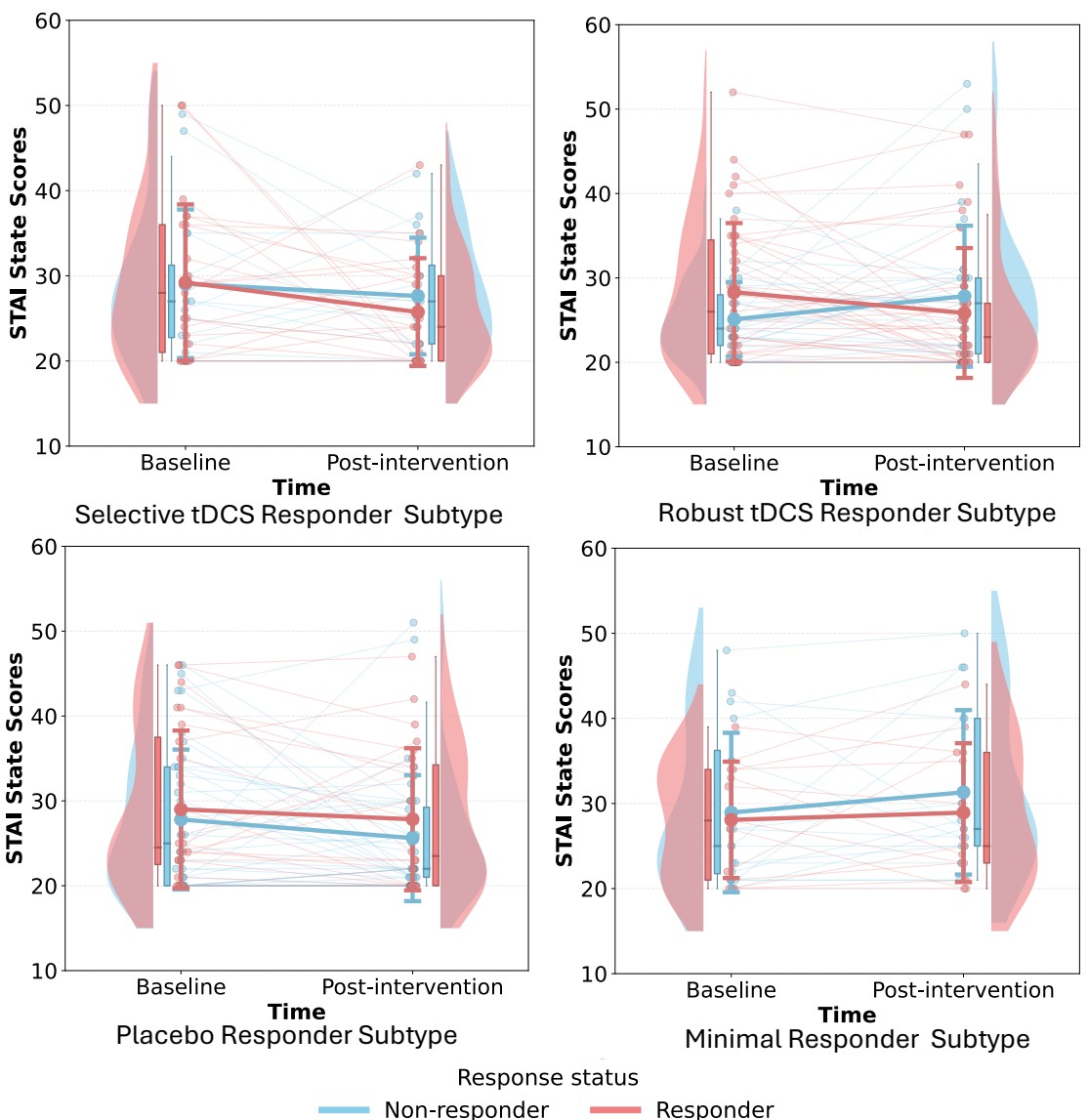

Figure S1: Pre–post changes in STAI-State scores across four clusters. Each panel shows baseline and post-intervention STAI-State scores for participants in the Selective tDCS Responder, Robust tDCS Responder, Placebo Responder, and Minimal Responder subtype clusters. Individual trajectories are displayed with light lines, overlaid with violin distributions and boxplots representing the score distributions at each time point. Colored trajectories indicate responder status (blue = non-responder; red = responder). Overall patterns highlight subtype-specific differences in anxiety reduction following the intervention.

**Selective tDCS Responder Subtype.** Individuals with moderate/severe baseline status show a very large improvement margin, with active tDCS producing gains more than ten

points greater than sham. Selective tDCS Responder subtype cluster exhibits the strongest clinical improvement. This clinical response aligns with the Selective tDCS Responder subtype's densely interconnected fronto-parietal and medial-frontal network architecture, characterized by strengthened links among superior frontal, angular, occipital, and precentral regions—an organization that appears especially receptive to modulation by tDCS.

**Robust tDCS Responder Subtype.** Individuals show clear and statistically significant gains under active tDCS compared with sham, including reliable improvement in the minimal and mild baseline groups. Robust tDCS Responder subtype cluster demonstrates a moderate but consistent benefit across individuals receiving active stimulation ($7.0 \pm 11.47$; $n = 5$). The observed pre–post reductions were consistent with the Robust tDCS Responder subtype's selectively strengthened fronto-parietal and frontal–temporal connections, a network configuration that may permit effective engagement by tDCS-driven neuromodulation.

**Placebo Responder Subtype.** Individuals demonstrate greater improvement during sham stimulation + cognitive training (CT) than active tDCS + CT, indicating sensitivity to CT rather than stimulation-dependent benefit. Placebo Responder subtype shows significant improvement under active tDCS ($7.3 \pm 6.95$; $n = 6$) and notable reductions even in the sham group ($12.8 \pm 13.00$; $n = 4$). Improvement in the absence of stimulation suggests strong expectancy or placebo effects.

**Minimal Responder Subtype.** Individuals exhibit little to no improvement under either sham or active tDCS, reflecting limited responsiveness across condition. Minimal Responder subtype cluster, in contrast, exhibits minimal clinical benefit.

### S7.1. Selective tDCS Responder Subtype

**Minimal/Mild Baseline Group.** Participants showed minimal pre–post improvement in both conditions, with no evidence of an Active benefit.

Table S5: Selective tDCS Responder Subtype (Minimal/Mild): Descriptive Statistics

| Condition | $n$ | Mean Change | SD |
|-----------|-----|-------------|------|
| Sham | 14 | $-0.50$ | 6.50 |
| Active | 20 | 0.10 | 5.56 |

Table S6: Selective tDCS Responder Subtype (Minimal/Mild): Statistical Tests

| Test | Statistic | $p$-value |
|------|-----------|-----------|
| Welch's $t$-test | $t = 0.28$ | 0.781 |
| Mann–Whitney $U$ | $U = 148$ | 0.805 |
| Cohen's $d$ | 0.07 | — |

**Moderate/Severe Baseline Group.**  Active participants showed a numerically larger reduction than Sham, but permutation testing indicated the effect was not statistically reliable.

Table S7: Selective tDCS Responder Subtype (Moderate/Severe): Descriptive and Permutation Results

| Condition | $n$ | Mean Change | SD | — |
|---|---|---|---|---|
| Sham | 2 | 7.00 | 2.83 | |
| Active | 3 | 18.00 | 7.00 | |
| Observed Difference (A–S) | | +11.00 | | |
| Permutation $p$-value | | 0.1956 | | |
| Cohen's $d$ | | 2.10 (unstable) | | |

**All Participants.**  When pooling Minimal/Mild and Moderate/Severe baseline groups, Active stimulation produced a slightly larger mean reduction in STAI-State scores than Sham in the Selective tDCS Responder subtype, but this difference was not statistically reliable. Both Welch's two-sample $t$-test and the Mann–Whitney $U$ test were nonsignificant, and the effect size was small.

Table S8: Selective tDCS Responder Subtype (All Participants): Active–Sham Statistical Tests

| Test | Statistic | $p$-value |
|---|---|---|
| Welch's $t$-test | $t = 0.69$ | 0.4968 |
| Mann–Whitney $U$ test | $U = 196$ | 0.7525 |
| Cohen's $d$ | 0.21 | — |

### S7.2. Robust tDCS Responder Subtype

**Minimal/Mild Baseline Group.**  This subtype showed the strongest and only statistically reliable Active–Sham difference among the Minimal/Mild participants. Active stimulation produced significantly greater symptom reduction than Sham across both parametric and nonparametric tests based on Mann-Whitney $U$ test.

Table S9: Robust tDCS Responder Subtype (Minimal/Mild): Descriptive Statistics

| Condition | $n$ | Mean Change | SD |
|---|---|---|---|
| Sham | 29 | $-2.38$ | 7.93 |
| Active | 34 | $-6.68$ | 9.57 |

**Moderate/Severe Baseline Group.**  Only Active participants were present ($n = 5$), so no condition comparisons could be conducted.

Table S10: Robust tDCS Responder Subtype (Minimal/Mild): Statistical Tests

| Test | Statistic | $p$-value |
|------|-----------|-----------|
| Welch's $t$-test | $t = 2.65$ | **0.0105** |
| **Mann–Whitney $U$** | $U = 649$ | **0.0309** |
| Cohen's $d$ | 0.59 | — |

**All Participants.** Across all participants in the Robust tDCS Responder subtype, Active stimulation produced a substantially larger mean reduction in STAI-State scores than Sham. This Active–Sham difference was statistically reliable in the Mann–Whitney $U$ test, with a medium-to-large effect size.

Table S11: Robust tDCS Responder Subtype (All Participants): Active–Sham Statistical Tests

| Test | Statistic | $p$-value |
|------|-----------|-----------|
| Welch's $t$-test | $t = 2.95$ | **0.0046** |
| **Mann–Whitney $U$** | $U = 753$ | **0.0197** |
| Cohen's $d$ | 0.73 | — |

### S7.3. Placebo Responder Subtype

**Minimal/Mild Baseline Group.** This subtype showed moderately larger improvement under Active stimulation, though results did not reach statistical significance.

Table S12: Placebo Responder Subtype (Minimal/Mild): Descriptive Statistics

| Condition | $n$ | Mean Change | SD |
|-----------|-----|-------------|-----|
| Sham | 32 | −3.25 | 6.75 |
| Active | 18 | −7.11 | 8.98 |

Table S13: Placebo Responder Subtype (Minimal/Mild): Statistical Tests

| Test | Statistic | $p$-value |
|------|-----------|-----------|
| Welch's $t$-test | $t = -0.90$ | 0.135 |
| Mann–Whitney $U$ | $U = 259$ | 0.129 |
| Cohen's $d$ | 0.52 | — |

**Moderate/Severe Baseline Group.** Active participants showed slightly smaller improvements than Sham, but permutation tests confirmed no reliable difference.

Table S14: Placebo Responder Subtype (Moderate/Severe): Descriptive and Permutation Results

| Condition | $n$ | Mean Change | SD |
|---|---|---|---|
| Sham | 4 | 12.75 | 12.99 |
| Active | 6 | 7.33 | 11.47 |
| Observed Difference (A–S) | | −5.42 | |
| Permutation $p$-value | | 0.3946 | |
| Cohen's $d$ | | −0.56 | |

**All Participants.** When Minimal/Mild and Moderate/Severe participants were combined, Active and Sham conditions produced very similar mean pre–post changes in STAI-State in the Placebo Responder subtype. Neither Welch's two-sample $t$-test nor the Mann–Whitney $U$ test showed a significant Active–Sham difference, and the overall effect size was small and slightly negative.

Table S15: Placebo Responder Subtype (All Participants): Active–Sham Statistical Tests

| Test | Statistic | $p$-value |
|---|---|---|
| Welch's $t$-test | $t = -0.49$ | 0.6249 |
| Mann–Whitney $U$ | $U = 426.5$ | 0.9395 |
| Cohen's $d$ | −0.12 | — |

### S7.4. Minimal Responder Subtype

**Minimal/Mild Baseline Group.** Participants in the Minimal Responder Subtype exhibited modest reductions in STAI-State scores in both Sham and Active conditions, with no statistically significant difference between stimulation types. Both Welch's $t$-test and Mann–Whitney $U$ test indicated nonsignificant effects.

Table S16: Minimal Responder Subtype (Minimal/Mild): Descriptive Statistics

| Condition | $n$ | Mean Change | SD |
|---|---|---|---|
| Sham | 12 | −4.00 | 8.47 |
| Active | 12 | −1.17 | 6.58 |

**Moderate/Severe Baseline Group.** Only Sham participants were present ($n = 3$, mean change = 5.33), so no Active–Sham comparison was possible.

**All Participants.** Across all participants in the Minimal Responder subtype, both Sham and Active conditions showed only modest pre–post changes, and Active stimulation did not outperform Sham. Welch's two-sample $t$-test and the Mann–Whitney $U$ test were nonsignificant, with a small effect size.

Table S17: Minimal Responder Subtype (Minimal/Mild): Statistical Tests

| Test | Statistic | $p$-value |
|---|---|---|
| Welch's $t$-test | $t = 0.92$ | 0.371 |
| Mann–Whitney $U$ | $U = 82$ | 0.581 |
| Cohen's $d$ | 0.34 | — |

Table S18: Minimal Responder Subtype (All Participants): Active–Sham Statistical Tests

| Test | Statistic | $p$-value |
|---|---|---|
| Welch's $t$-test | $t = 0.54$ | 0.5960 |
| Mann–Whitney $U$ | $U = 117$ | 0.5822 |
| Cohen's $d$ | 0.19 | — |

## S7.5. Demographic and Psychological Profiles of Four Clusters

We identified four clusters reflecting distinct demographic and psychological profiles (Table S19). Based on prior work showing strong depression–anxiety comorbidity (Kessler et al., 2008; Lamers et al., 2011), we included BDI-II subscales to capture shared affective variance that may influence connectivity patterns. Two clear demographic and psychological profiles patterns emerged among the four clusters. The Minimal Responder and Selective tDCS Responder subtypes cluster groups had more female participants and showed higher baseline STAI and BDI-II scores. In contrast, the Robust tDCS Responder and Placebo Responder subtypes were the groups with largest participant numbers, with moderate symptoms and balanced sex ratios.

Table S19: Demographic, clinical, and connectivity characteristics across the four data-driven clusters. $n$ = number of participants; $SD$ = standard deviation. BDI-II subscales include: Somatic (physical/vegetative symptoms), Affective (emotional symptoms), and Cognitive (self-critical or distorted thinking patterns).

| Measure | Selective tDCS Responder Subtype | Robust tDCS Responder Subtype | Placebo Responder Subtype | Minimal Responder Subtype |
|---|---|---|---|---|
| n | 39 | 68 | 60 | 29 |
| Proportion (%) | 19.60 | 34.17 | 30.15 | 14.57 |
| Age (Mean (SD)) | 70.49 (3.46) | 70.24 (4.25) | 72.30 (4.81) | 72.62 (5.43) |
| STAI-State Pre (Mean (SD)) | 29.15 (8.87) | 26.94 (6.98) | 28.28 (8.62) | 28.55 (8.20) |
| STAI-Trait (Mean (SD)) | 28.56 (8.41) | 27.91 (6.28) | 28.37 (7.57) | 29.07 (7.21) |
| BDI Total (Mean (SD)) | 2.59 (3.23) | 2.76 (3.76) | 3.65 (4.08) | 4.72 (4.17) |
| BDI Somatic (Mean (SD)) | 1.64 (1.90) | 1.78 (2.15) | 2.38 (2.54) | 3.10 (2.64) |
| BDI Affective (Mean (SD)) | 0.38 (0.75) | 0.44 (0.84) | 0.55 (0.98) | 0.79 (1.01) |
| BDI Cognitive (Mean (SD)) | 0.56 (1.21) | 0.54 (1.51) | 0.72 (1.40) | 0.83 (1.26) |
| Male (n) | 10 | 23 | 22 | 6 |
| Female (n) | 29 | 45 | 38 | 23 |
| Male (%) | 25.64 | 33.82 | 36.67 | 20.69 |
| Female (%) | 74.36 | 66.18 | 63.33 | 79.31 |
| Mean Red-Box Connectivity | 0.3241 | 0.2889 | 0.2065 | 0.2365 |

