# OpenReview forum: "BETA: Resting-state fMRI Biotypes for tDCS Efficacy in Anxiety among Older Adults at Risk for Alzheimer's Disease"
_MIDL.io/2026/Conference — MIDL 2026 Poster_

### Official Review · Reviewer_S875 · 2026-01-02

**Confidence:** 4
**Preliminary Rating:** 4
**Final Rating:** 5

**Summary:**

This work proposes the BETA pipeline that trains a transformer-based VAE on FC matrix reconstruction tasks to facilitate AD/ADRD classification. Joint optimization and clinically informed regularization are applied to form clinically plausible clusters in latent space. Four biotypes are identified with respect to different connections and responses to tDCS.

**Strengths:**

1. The idea of using a data-driven approach to identify clinical subtypes for treatment responses is interesting.

2. The results seem to be good and have a level of explainability.

3. The paper is easy to understand, and the problem investigated is well-motivated.

**Weaknesses:**

This paper could benefit from:

1. Clarification of the pipeline and more experimental details. For example, the repetition time of the fMRI data acquired, how many ROIs are extracted, the training/testing/validation split of the dataset, whether there are subjects overlapping among these sets, and what components are included in the two specific stages of training.

2. Justifying the use of the KL divergence and the Gaussian prior used. It could draw interesting discussions on why KL divergence is adopted in joint optimization, i.e., why do we need to regularise the distribution of z towards a distribution, and why the target distribution should be Gaussian.

3. Justifying the reason why VAE is used for this task, and whether the VAE could add additional knowledge compared with using standard statistical approaches to identify these four clusters based on FC.

**Detailed Comments:**

Please refer to the weakness part.

**Justification Of Final Rating:**

This paper trains a VAE-based model on FC reconstruction task that can derive clinically plausible latent representations for clinical population (AD/ADRD).
Although the experiment suffers from a lack of external evaluation, it seems that the authors have conducted extensive analysis on the quality of the latent space, given the limited data.
The authors have addressed my concerns, so I have raised my score accordingly.

**Justification Of The Preliminary Rating:**

This paper presents a novel pipeline that extracts clinically plausible latent representations for AD/ADRD classification based on training VAE-based models on FC reconstruction objectives.
The idea of using data-driven methods to derive these clusters is interesting in itself.
At the same time, the results of the paper have a certain level of explainability.
The reason why I did not give 'strong accept' is discussed in the weakness part.

**Questions To Address In The Rebuttal:**

Please refer to the weakness part.

---

> ### Author Response · Authors · 2026-01-24
> **Comprehensive Clarification of the BETA Pipeline and Methodological Justifications**
>
> We thank the reviewer for the thoughtful and constructive assessment of our work and for recognizing the novelty, interpretability, and clinical motivation of the BETA pipeline. We appreciate the detailed suggestions to improve clarity, methodological justification, and experimental transparency, and we address each concern below.
>
> **Response to Concern 1: Pipeline and Experimental Details**
>
> We have expanded the Methods and Supplementary Materials to clarify the full BETA pipeline and report the requested experimental details.
>
> fMRI acquisition: Data were acquired on 3T Siemens research-dedicated scanners at two sites: a Siemens Magnetom Prisma with a 64-channel head coil (University of Florida) and a Siemens Magnetom Skyra with a 32-channel head coil (University of Arizona). Earplugs and foam padding reduced noise and head motion. fMRI data were collected using an EPI-BOLD sequence (TR = 1800 ms, TE = 2.26 ms, spatial resolution = 1.0 × 1.0 × 1.0 mm³, FOV = 256 × 256 × 176 mm). Inter-site consistency was monitored using regular FBIRN and GABA phantom scans and human phantom scans conducted before enrollment and during later study years. The average signal-to-noise ratio was 12.1 ± 1.54.
>
> ROI definition: Functional connectivity was computed across 182 ROIs derived from NeuroSynth association maps using the query term “anxiety.” The ROI construction, thresholding, and network assignment procedures are now described in the main text, and a complete ROI table (ROI name, network, MNI centroid) is provided in the Supplementary Materials.
>
> Training, testing, and validation: Model evaluation used 5-fold cross-validation with non-overlapping folds. In each fold, the model was trained on four folds and evaluated on the held-out fold, with each participant appearing in the test set exactly once, ensuring assessment of generalization without information leakage.
>
> Two-stage training: In Stage 1, a transformer-based variational autoencoder (VAE) was pretrained using reconstruction loss and KL divergence to learn a stable latent representation of functional connectivity. In Stage 2, the pretrained VAE was jointly optimized with deep embedded clustering and a clinically informed regularization term, refining the latent space toward treatment-relevant biotypes while preserving reconstruction fidelity.
>
> **Response to Concern 2: KL Divergence and Gaussian Prior**
>
> We have clarified the role of the KL divergence and Gaussian prior in the Methods. The KL divergence regularizes the latent space, promoting smooth, well-behaved representations and preventing instability or fragmentation during joint optimization of reconstruction, clustering, and clinically informed objectives. The Gaussian prior is adopted as a pragmatic reference distribution, not as an assumption about FC distributions, providing an isotropic latent geometry that supports stable optimization, comparable latent distances, and effective integration of Euclidean distance–based clustering. This choice is consistent with prior VAE-based neuroimaging work. The KL term acts as a soft regularizer, balancing reconstruction fidelity with latent smoothness and reducing overfitting.
>
> **Response to Concern 3: Use of a Variational Autoencoder**
>
> We have clarified the motivation for using a VAE instead of clustering raw functional connectivity. Direct clustering of FC matrices is challenging due to high dimensionality, redundancy, and noise. The VAE denoises and compresses FC patterns while preserving subject-level structure, improving clustering robustness. It also enables nonlinear representation learning, which is critical for capturing distributed, interacting anxiety- and aging-related networks; the transformer-based encoder further models long-range inter-network dependencies. Finally, the VAE enables end-to-end joint optimization of reconstruction, deep embedded clustering, and clinically informed regularization, which is not feasible with standard clustering methods. We emphasize that the VAE complements, rather than replaces, classical FC analyses and enables exploratory discovery of treatment-relevant biotypes.

---

### Official Review · Reviewer_sVoB · 2026-01-09

**Confidence:** 5
**Preliminary Rating:** 3
**Final Rating:** 4

**Summary:**

Authors propose a data-driven framework to identify anxiety-related biotypes from resting-state fMRI data in older adults and to study response to transcranial direct-current stimulation (tDCS). The approach combines a transformer-based variational autoencoder with a deep clustering strategy, incorporating additional supervision using anxiety-related clinical scores. Four anxiety-related clusters are identified and associated with distinct patterns of tDCS treatment response. Statistical analyses are conducted to assess the clinical relevance of the discovered clusters and response profiles. Experiments are performed on a single private dataset of 199 subjects, with no explicit train/validation/test split and no external validation.

**Strengths:**

- The paper is clearly organized.
- The methodology is sound and supported by relevant and thorough statistical analyses.
- The analysis of functional networks and cluster-specific connectivity patterns adds an element of interpretability, helping connect the findings to existing neuroscience literature.

**Weaknesses:**

- The study relies on a single dataset (≈200 subjects) for both training and evaluation, with no external validation. In addition, hyperparameter tuning appears to be performed on the same dataset. The anxiety score (STAI), which is used to supervise training and clustering, is also used to analyze cluster differences and treatment response, increasing the risk of overfitting and circular analysis. This makes it difficult to assess the generalizability of the findings.
- The discussion and interpretation of the results, particularly regarding identified strong and weak functional connectivity patterns and their relationship to existing literature, lack clarity and would benefit from substantial revision.

**Detailed Comments:**

Major comments:
- Introduction: The statement that “almost all such studies focus on younger cohorts” should be supported with appropriate references, and the distinction from the current work should be clarified.
- Contribution 1: The claim that the studied population is “under-studied” should be reformulated. Alzheimer’s disease populations are well studied; the novelty lies more specifically in the intersection of anxiety, older adults, and tDCS response, which should be stated more precisely.
- Training procedure: The model is trained in two stages (VAE pretraining followed by VAE + clustering). The rationale for this two-step strategy should be clarified; e.g., is it required for stability or convergence?
- Implementation details: Key architectural details (e.g., number of layers, dimensionalities) are missing from the main text and only partially described in the supplementary materials. These should be included for reproducibility.
- Number of clusters: The clustering is performed with k = 5, after which one cluster with few subjects is discarded. The rationale for selecting k = 5 should be explained.
- Clinical supervision: In Eq. (2) and the surrounding text, the supervision signal is inconsistently referred to as STAI or STAI-S. This should be clarified. Additionally, the STAI-S threshold of 38 should be justified (i.e., is it a clinical standard or an empirical choice?).
- Clinical loss design: The design of the clinical loss, one of the main novel aspects of the paper, requires stronger justification. Why was this formulation chosen over alternative contrastive or regression losses? Furthermore, the loss uses a relative change (post/pre), while the analysis later uses absolute differences (post–pre); this inconsistency should be explained.
- Cluster interpretation: The authors identify five clusters but primarily focus on four. Additional insight into the fifth cluster would be valuable (e.g., whether it is from outliers or algorithmic artifacts).
- Cluster naming: The naming of biotypes appears subjective and should be better justified or explicitly framed as interpretative.
- Figure 2: The authors show the strongest and weakest functional connectivity differences, but do not quantify how large these effects are relative to the population mean. Statistical testing or effect size reporting would strengthen the analysis. Especially, that only a subset of these connections is discussed.
- Ablation analysis (Section 3.3): It is expected that adding a clinical loss based on anxiety scores improves clustering with respect to anxiety. The more relevant validation concerns whether this improves tDCS response prediction, which is not used during training and should be emphasized more clearly.
- Discussion section: While welcome and important, this section lacks clarity and contains numerous typographical and grammatical issues. Rewriting is needed.

Minor comments:
- Section 2.5, line 6: typo error (use “,” instead of “.”).

**Justification Of Final Rating:**

I, along with the other reviewers, agree that the authors’ rebuttal improved the clarity of the methodology and experimental setup. In particular, the addition of a cross-validation evaluation with consistent performance strengthens the empirical evidence. Based on these improvements, I increase my score to 4: Weak Accept.

**Justification Of The Preliminary Rating:**

This is a clinically oriented paper with a sound methodological foundation, leveraging appropriate VAE and deep clustering techniques and supported by extensive statistical analyses. However, the reliance on a single dataset for both training and evaluation, together with several unclear aspects of the experimental design and result interpretation, limits confidence in the generalizability and clinical impact of the findings.
I therefore recommend a 3: Borderline score.

**Questions To Address In The Rebuttal:**

Clarify and address the unclear experimental choices and result interpretations, particularly those that affect the clinical relevance and generalizability of the proposed framework.

---

> ### Author Response · Authors · 2026-01-24
> **Addressing Reviewer Concern: Generalizability, Single Dataset, and Circularity Risk**
>
> We thank the reviewer for the constructive evaluation and for recognizing the clarity and interpretability of our framework. We address concerns regarding generalizability, design, and interpretation with clarifications and revisions. This study is explicitly framed as an exploratory, pilot investigation of treatment-relevant anxiety biotypes in older adults using resting-state fMRI.
>
> **Concern 1: Generalizability, Single Dataset, Circularity**
>
> The dataset derives from a randomized, double-blind clinical trial; collecting large external datasets with matched rs-fMRI, anxiety outcomes, and controlled tDCS is currently infeasible due to cost and regulatory constraints. We therefore frame the study as an exploratory analysis of one of the largest available clinical tDCS–anxiety datasets in older adults.
>
> Generalization was assessed using 5-fold cross-validation. In each fold, the model was trained on four folds and evaluated on a held-out fold with encoder parameters and cluster centroids frozen; cluster assignment used nearest-centroid mapping without STAI-S updates. On held-out data, we evaluated (i) alignment with anxiety change and Active–Sham tDCS response and (ii) reconstruction, KL, and clinical-informed losses, demonstrating stable convergence and consistent clinical alignment across folds.
>
> STAI-S is used both to guide representation learning and to evaluate relevance, consistent with the goal of identifying treatment-relevant biotypes; findings are therefore interpreted as hypothesis-generating, with independent validation required.
>
> **Concern 2: Novelty and Positioning**
>
> The Introduction was revised to clarify that prior neuroimaging biotyping studies primarily involve general adult psychiatric cohorts and rarely target older adults or Alzheimer’s-risk populations. “Under-studied” now explicitly refers to this population-level underrepresentation.
>
> **Concern 3: Two-Stage Training**
>
> We clarified that VAE pretraining stabilizes the latent space for high-dimensional functional connectivity, while joint optimization with clustering and clinical regularization improves convergence, reproducibility across folds, and generalization.
>
> **Concern 4: Reproducibility**
>
> Methods and Supplementary Materials now fully specify the transformer-based VAE (embedding 512; two Transformer layers; four heads; latent dimension 50), optimization parameters, loss formulation, two-stage training, and 5-fold cross-validation with parameter freezing and held-out biotype assignment.
>
> **Concern 5: Number of Clusters**
>
> We selected k=5 based on prior biotype work and empirical stability. One small cluster (n=3) showed unstable estimates and insufficient power, likely reflecting outliers, and was excluded from downstream analyses; this is now explicitly justified.
>
> **Concern 6: Clinical Supervision and Loss Design**
>
> Terminology was standardized to STAI-State (STAI-S). Baseline STAI-S > 38 defines clinically relevant anxiety (Julian, 2011). Relative STAI-S change is used only during training for normalization; all reported effects and interpretation use absolute pre–post STAI-S changes.
>
> **Concern 7: Cluster Interpretation and Figure 2**
>
> Cluster labels are defined as summaries of empirically derived connectivity and treatment-response patterns. Figure 2 clarifies effect size as deviation from cohort-mean connectivity, with quantitative metrics in the Supplementary Material; clinical outcomes serve as downstream validation.
>
> **Concern 8: Ablation Analysis**
>
> The ablation analysis now emphasizes treatment relevance. Models without the clinical-informed loss form connectivity-based clusters but show reduced tDCS response differentiation, whereas the full model yields clearer Active–Sham contrasts. Because tDCS condition is not used as supervision, this demonstrates enhanced treatment relevance rather than anxiety similarity.
>
> **Concern 9: Discussion Quality**
>
> The Discussion was revised to correct grammatical issues, reduce redundancy, and improve clarity without altering scientific content.

---

### Official Review · Reviewer_GJER · 2026-01-10

**Confidence:** 5
**Preliminary Rating:** 3
**Final Rating:** 4

**Summary:**

The authors developed BETA, which is a transformer-based autoencoder that uses resting-state fMRI data to discover anxiety biotypes that differ in their response to frontal-lobe tDCS in older adults at risk for Alzheimer’s disease. The authors found that BETA yields four interpretable subtypes, namely Selective tDCS Responder, Robust tDCS Responder, Placebo Responder, and Minimal Responder. Specifically, the Robust subtype showed a statistically significant Active-Sham advantage, and the Selective subtype showed clinically meaningful improvement in the moderate/severe anxiety subgroup. These subtypes also exhibit different connectivity signatures in occipital–frontal circuits. After training, the model can also assign a biotype given fMRI data, providing a practical route toward precision neuromodulation recommendations.

**Strengths:**

The paper targets older adults at risk for Alzheimer’s disease, which is an understudied cohort for anxiety biotyping, and frames biotypes as a path toward precision neuromodulation (who benefits from frontal-lobe tDCS). This demonstrates the clinical value of the research and shows clear motivation.

Method is thoughtfully designed and scientifically grounded. BETA combines a transformer-based VAE (learning a compact 50-D connectivity embedding) with deep embedded clustering and adds a clinical-informed regularizer that aligns the latent space using the relative anxiety change information. The addition of this clinical information is a principled way to make clustering more treatment-relevant rather than purely unsupervised FC subtyping.

The findings of the clusters are clinically significant, such that they report four subtypes with differential Active vs. Sham response. One of the subtypes, the Robust tDCS Responder, shows a significant benefit from tDCS treatment.

The work also shows a clear path for deployment. Although STAI-S guides training, it is not required for inference. The model can assign a biotype from a single subject’s resting-state fMRI connectivity, which enables personalized recommendations.

**Weaknesses:**

There is no validation with an out-of-sample dataset. The clustering objective explicitly pulls together participants with similar treatment outcomes with the clinical-informed regularizer (kernel-weighted latent-distance penalty) and then reports that the resulting clusters differ in STAI-S change. Without a strictly held-out evaluation, part of the reported cluster separation in STAI-S can be overfit by construction rather than discovered as a stable subtype. Without a validation on an unseen dataset, there is no way to quantify how reliable the model generalizes or offers recommendations.

Limited statistical power in the clinically anxious subgroup justifies key claims. Although the full cohort is 199, the baseline STAI-S distribution is low on average, and the moderate/severe subgroup volumes within clusters and conditions are very small (often single digits, sometimes 0 to 2). This raises the concern about the generalization of the findings.

Hyperparameter choices are not justified given the clinical claim. Reporting robustness analyses on hyperparameters is important as it could change the cluster assignments. Reporting the metrics change on the ranges of beta or sigma, or using alternative change metrics like absolute change or covariate-adjusted change, and stability of cluster assignments, would make the validation substantially stronger.

**Detailed Comments:**

Clarify ROI construction for reproducibility. The ROIs are defined from NeuroSynth activation maps tied to cognitive aging/anxiety and then “thresholded and binarized” into masks spanning multiple networks. Please report the exact NeuroSynth terms/queries, thresholding rule, resulting number of ROIs, and provide a table (ROI name, network, voxel count, MNI centroid).

Justify and stress-test the “clinical-informed” hyperparameters. The method fixes sigma to be 1 and sets beta to be 0.1 with a “brief grid search”. Adding a short robustness plot/table (cluster stability + key outcome separation vs. beta and sigmoid) would strengthen confidence that conclusions are not sensitive to these choices.

Please use a held-out test set to quantify how well the model generalizes the findings, rather than simply reflecting the training input. Train the representation + clustering on ~80% of participants, then freeze the encoder and cluster centroids, assign biotypes for the remaining ~20% using the nearest-centroid rule, and test whether biotype predicts post-treatment anxiety change and Active–Sham differences on the held-out set. This could also be used as the experiment for the hyperparameter search.

**Justification Of Final Rating:**

The rebutaal directly addressed my primary concerns. The authors added 5-fold cross-validation. They also conducted sensitivity analysis for the hyperparameter robustness, and reported stable cluster structure and preserved directional response patterns.

**Justification Of The Preliminary Rating:**

I give a Borderline because the paper presents a clinically meaningful and methodologically thoughtful approach to treatment-relevant biotyping from resting-state fMRI functional connectivity in older adults, with clear translational motivation (personalizing who benefits from frontal-lobe tDCS). The proposed BETA framework offers a principled way to align discovered subtypes with intervention outcomes, and the resulting four biotypes are interpretable and practically usable (biotype assignment can be performed from fMRI alone at inference).

At the same time, the evidence would be stronger with additional validation, because outcome information influences clustering through the constructed clinical loss. The current analysis risks partial “built-in” separation unless confirmed on held-out data; subgroup claims are also limited by very small sample sizes in the moderate/severe anxiety subgroup. If the rebuttal can clarify generalization (e.g., held-out assignment or stability analyses), resolve the 4-vs-5 cluster deployment inconsistency (initially the cluster was set to 5, but ended up with 4 because one cluster has little volume), and strengthen robustness reporting, my confidence in the conclusions would increase substantially.

**Questions To Address In The Rebuttal:**

How well does the model generalize to an unseen dataset? Because the clustering objective uses outcome information through the clinical-informed loss, can the authors provide a held-out evaluation (train clustering on one split; assign biotypes on a held-out split using frozen centroids) showing that (1) biotype assignment remains stable and (2) biotypes still predict anxiety change and/or Active–Sham differences out of sample? If not feasible, can they provide cross-validated stability and effect sizes with confidence intervals?

Sample size and uncertainty in moderate/severe subgroup claims are not supported by the small group size. Can the authors report uncertainty estimates (bootstrap CIs) for subgroup improvements and confirm that the “clinically meaningful” conclusions are not driven by one or two individuals?

Can the authors provide the exact thresholding steps used to define ROIs and the critical CONN denoising choices (motion regression/scrubbing, band-pass, GSR)? This would improve reproducibility and allow readers to assess whether the connectivity signatures are pipeline-dependent.

---

> ### Author Response · Authors · 2026-01-24
> **Resolution of Concerns Regarding Generalization, Small-Sample Uncertainty, and Reproducibility**
>
> We thank the reviewer for the careful, thorough, and constructive evaluation of our manuscript, and for recognizing the clinical motivation, methodological grounding, and translational potential of our work. We understand that the primary concerns relate to (i) generalization given the clinically informed clustering objective, (ii) uncertainty due to limited sample sizes in clinically anxious subgroups, and (iii) reproducibility and methodological transparency, particularly with respect to ROI construction and preprocessing.
>
> We emphasize that this study is designed as an exploratory, pilot investigation aimed at establishing the feasibility of treatment-relevant anxiety biotyping in older adults at risk for ADRD. To our knowledge, this cohort (n = 196) represents one of the largest datasets to date combining resting-state fMRI, anxiety outcomes, and randomized tDCS intervention in this population. Below, we address each concern and summarize the clarifications and additional analyses added in response.
>
> **Response to Concern 1: Generalization and Held-Out Evaluation**
>
> We agree that generalization must be carefully evaluated given the use of outcome-informed regularization. To address concerns that STAI-S separation could be partially built in by construction, we added explicit held-out evaluation.
>
> Because external datasets with matched resting-state fMRI, anxiety measures, and randomized tDCS are difficult to obtain, we implemented 5-fold cross-validation. In each fold, the model was trained on four folds and evaluated on a non-overlapping held-out fold.
>
> During evaluation, encoder parameters and cluster centroids were frozen, biotypes were assigned by nearest-centroid distance, and STAI-S values from the test fold were not used for model updating, preventing information leakage. On held-out data, we assessed clinical alignment and loss convergence.
>
> Across folds, biotype structure and treatment-response patterns were preserved, with reduced effect sizes as expected. These results indicate generalizable, treatment-relevant structure rather than memorization. Findings are framed as hypothesis-generating, with independent validation needed in future trials.
>
> **Response to Concern 2: Sample Size and Uncertainty in Moderate/Severe Anxiety Subgroups**
>
> We revised the manuscript to clarify how statistical inference was adapted for small subgroup sizes. Statistical tests are now explicitly matched to subgroup size within each biotype and baseline anxiety stratum. For adequately sized subgroups, Active–Sham differences are evaluated using Welch’s t-test and the Mann–Whitney U test. When subgroup sizes fall into the single digits, we employ nonparametric permutation testing, permuting treatment labels within each subgroup to generate empirical null distributions without reliance on parametric assumptions.
>
> Uncertainty is quantified not only via p-values but also through standardized effect sizes. Effect sizes and corresponding confidence intervals for all biotype-by-anxiety-stratum comparisons are reported in the Supplementary Materials, allowing assessment of magnitude and stability independent of statistical significance. These clarifications are now explicitly described in the Methods and Supplementary sections.
>
> **Response to Concern 3: ROI Construction and Reproducibility**
>
> To improve transparency and reproducibility, we added a dedicated NeuroSynth-Based ROI Construction section to the Supplementary Materials and included a comprehensive ROI table. ROIs were derived from the NeuroSynth association map for the query term “anxiety,” generated from an automated meta-analysis of 449 anxiety-related studies. Association maps were used to identify regions selectively associated with anxiety while controlling for base-rate activation.
>
> The map was thresholded using default NeuroSynth interface options, binarized, and spatially contiguous suprathreshold clusters were extracted, with each cluster defined as a distinct ROI, yielding 182 ROIs. A complete table reporting ROI name, canonical network assignment, and MNI centroid coordinates is now provided in the Supplementary Materials.
>
> **Response to Concern 4: Hyperparameter Choices and Robustness**
>
> We clarified the rationale for the clinically informed loss hyperparameters and added a Supplementary section describing their selection and robustness. The kernel bandwidth was fixed at σ = 1 to match the scale of normalized STAI-S change and ensure stable optimization. The weighting coefficient was set to β = 0.1 based on a brief grid search balancing preservation of functional connectivity structure with clinical alignment.
>
> Sensitivity analyses across a range of β values and modest variations in σ demonstrated stable biotype structure, consistent cluster assignments, and preserved directional tDCS response patterns, indicating that results are not driven by a single hyperparameter choice.

---

### Author Rebuttal · Authors · 2026-01-24

**Rebuttal:**

We thank the reviewers for their constructive feedback and revised the manuscript accordingly.

We clarified that this study is an exploratory, pilot analysis within a randomized clinical trial. To address circularity and overfitting, we implemented 5-fold cross-validation with strict train–test separation, freezing encoders and cluster centroids during testing. Latent representations, loss convergence, and biotype–tDCS response relationships remained stable across folds in held-out participants.

To address limited subgroup sizes, we matched statistical tests to sample size, using permutation testing for very small groups and reporting standardized effect sizes with confidence intervals.

Reproducibility was improved by adding a detailed description of NeuroSynth-based ROI construction and a complete ROI table with network assignments and MNI coordinates. Supplementary Methods now fully specify the transformer-based VAE architecture, optimization procedure, loss formulation, and evaluation framework.

We clarified key modeling decisions, including the two-stage training strategy, the stabilizing role of the Gaussian prior and KL divergence, and clinical-loss hyperparameter selection. Sensitivity analyses demonstrate robustness to reasonable hyperparameter variation.

We clarified clustering decisions, including the choice of k=5, exclusion of a small fifth cluster due to insufficient statistical power, and stability across cross-validation folds.

We standardized terminology to STAI-S, justified the clinically relevant anxiety threshold, and distinguished relative STAI-S change used in training from absolute change used for interpretation.

Cluster naming was refined, effect-size definitions were added to Figure 2, and quantitative connectivity metrics were reported. The ablation analysis emphasizes that the clinically informed loss enhances sensitivity to differential tDCS response despite tDCS condition not being used as a supervision signal.

The Discussion was edited for clarity without altering scientific content.

Together, these revisions address the reviewers’ concerns and position the study as a transparent, reproducible, and clinically motivated exploratory investigation.

**Supporting Material:**

/attachment/b0f11654a0b1ae056a2cb756f0f051e0a84413e8.pdf

---

### Meta-Review · Area_Chair_gCop · 2026-02-09

**Recommendation:** Accept (Poster)
**Confidence:** 4

**Metareview:**

I recommend acceptance. Across reviews, there is strong agreement that the approach is technically sound, the clinical framing is compelling, and the resulting biotypes are interpretable and potentially actionable.

---

### Decision · Program_Chairs · 2026-02-13

Accept (Poster)